# Biomechanical tradeoffs in stroller running: Reduced vertical impact loading and increased torsional injury risk

Joseph M. Mahoney[1,2,3☯], Amy Lista[2,4¶], Diego Carbajal[2¶], Benjamin W. Infantolino[2¶], Allison R. Altman-Singles[1,2☯]*

1 Mechanical Engineering, The Pennsylvania State University, Berks College, Reading, Pennsylvania, United States of America, 2 Kinesiology, The Pennsylvania State University, Berks College, Reading, Pennsylvania, United States of America, 3 Mechanical Engineering, Alvernia University, Reading, Pennsylvania, United States of America, 4 Occupational Therapy, Alvernia University, Reading, Pennsylvania, United States of America

☯ These authors contributed equally to this work.
¶ These authors contributed equally to this work.
* ara5093@psu.edu

## Abstract

This study evaluated how running with a stroller influences biomechanical parameters commonly associated with injury risk. Specifically, the study investigated changes in vertical and torsional loading and tibial acceleration in runners pushing a jogging stroller compared to running without one. Thirty-eight healthy adult runners participated in trials where they ran over a force plate with and without a stroller. Key measurements included vertical and anterior-posterior ground reaction forces, tibial accelerations, and free moments. The findings demonstrated significant reductions in vertical loading metrics, including vertical impact peak, vertical instantaneous and average loading rates, and vertical impulse, by 8–17% when running with a stroller. These reductions suggest a decreased risk of overuse injuries commonly associated with vertical forces. Conversely, torsional loading parameters, such as peak free moment and free moment impulse, increased significantly, with some measures rising by more than 400%. This increase in torsional loading indicates an elevated risk of stress-related injuries, particularly tibial stress fractures. In addition, tibial acceleration decreased slightly, though to a lesser extent than vertical loading metrics. These results highlight a biomechanical tradeoff when running with a stroller. While the reduction in vertical loading may mitigate the risk of bone stress and overuse injuries, the simultaneous increase in torsional loading could heighten the likelihood of torsional stress injuries. The study underscores the need for further research to explore mitigation strategies, such as optimized stroller designs or alternative pushing techniques, to balance these risks. The findings contribute valuable insights for runners, coaches, and stroller manufacturers aiming to promote safer running practices for caregivers using jogging strollers.

**Data availability statement:** All Microsoft Excel files of data in figures will be provided via public database (OSF) upon acceptance. Data can be accessed here: https://osf.io/g4v3w/.

**Funding:** The authors received institutional funding from Pennsylvania State University (to ARA) and Alvernia University (to JMM) to support this work.

**Competing interests:** The authors have declared that no competing interests exist.

## Introduction

Running is a common form of physical activity that provides benefits for both mental and physical health. Up to 79% of runners are injured each year, which disrupts healthy running routines [1]. The causes of running overuse injuries are multifactorial [2]. Gait mechanics play an important role in modulating the risk of overuse injuries [3–6]. Additionally, inconsistent participation due to lower running volume [7] or frequent fluctuations [8] can increase the risk of overuse injuries.

Caregivers of young children often face physiological and logistical challenges that affect their running directly and indirectly. For runners, this can lead to disruptions in routines as they figure out how to exercise with a young baby. Many runners choose to resume running with a running stroller – defined here as a three-wheeled stroller with two large rear and one large front wheel that is directionally locked. The running stroller allows for some degree of shock absorption and easy navigation on flat, straight paths. While the design of a running stroller is intended to minimize drag, making the stroller easy to push, the increasing weight of the stroller and child, the height of the handlebars, and the shortened field for anterior foot placement present challenges that may require biomechanical gait adjustments. It is unknown whether these gait adjustments may increase the risk of overuse injury in this vulnerable population.

With the stroller industry projected to nearly double between 2024 and 2033 [9], it is crucial to better understand how the stroller may affect running mechanics, as well as potential injury risk. Previous work has shown that running with a stroller resulted in increased exertion versus running without a stroller [10–12]. It is further noted in the literature that walking with a double stroller [13] or running with a stroller uphill [14] further increases exertion compared to a single stroller on flat ground. Some studies have shown that running with a stroller resulted in decreased stride length [10,15], while others found that changes in speed confounded the ability to detect changes in stride length [12,15,16]. Only one study has examined joint kinematics while running with a stroller and found increased anterior trunk lean, pelvic tilt, and hip flexion, as well as decreased trunk rotation [16].

Kinematic changes along the lower-extremity planes suggest that running with a stroller affects the gait of a runner due to the body being more flexed while running [16]. However, previous stroller research has not investigated *kinetics*. It was expected that changes in kinematics are reflective of changes in kinetics. As an individual bout of running is comprised of thousands of these impacts, the repetitive loading can lead to the accumulation of microdamage, which, over time, can lead to overuse injuries. Ground reaction force and tibial acceleration reflect how hard the body contacts the ground with each step. These mechanical ground reaction force factors include high-impact loading [5,15,16], vertical impulse (VIMP) [17], or high torsional forces [4] which are associated with increased overuse injury risk. Impact loading is related to the initial contact with the ground in the vertical plane. Previous studies have shown that impact loading is characterized by a high vertical impact peak (VIP), high vertical average (VALR), and instantaneous loading rates (VILR). These impact parameters can be altered by changes in footstrike mechanics [17] or

stride length [18]. High tibial acceleration can contribute to both the risk of injury while running [19] and peak acceleration of the tibia (TS) can serve as a surrogate measure of these high impact load vertical ground reaction force (vGRF) metrics [20].

High torsional moments are characterized using the relative moment in the plane of the floor, normalized by the center of pressure, termed the free moment. Key parameters include the peak absolute free moment ($|FM|_{max}$), peak adduction free moment (FMADD), the free moment at peak braking force (FMBRAK), and the overall free moment impulse (FMIMP) [4]. High free moment parameters have been identified as risk factors for tibial stress fractures [4]. In addition, high torsion moments occur with an increase in load carriage [21]. There is a potential relationship between high impact loading and high free moment [22], as motion control shoes are implemented, both can be reduced.

Given the unknown impact of stroller running on injury risk, this study aimed to determine if running with a stroller changed common metrics associated with injury risk, specifically impact loading, torsional loading, and tibial acceleration. It was expected that running with a stroller may decrease metrics of vertical loading due to the shortened stride length and potential load mitigation through the arms on the handlebars. In contrast, torsional loading was expected to increase while running with a stroller due to the increased functional weight of the stroller-runner complex. It is important to investigate these mechanical contributors to injury by collecting running data with and without a stroller across a force plate to better assess the safety of running with a stroller. These findings may better inform stroller design, coaching strategies, as well as injury prevention and rehabilitation protocols within the stroller-running population. The current research revealed reduced vertical impact parameters, tibial acceleration, and increased free moment parameters, as was hypothesized.

## Methods

### Subjects

Healthy male and female runners were recruited from June 20, 2022 until April 22, 2024 for this study through running clubs, teams, and word of mouth. Participants were 18–45 years old and must have regularly run a minimum of 5 miles per week. Participants were free from any injuries and cardiovascular disorders that would affect their ability to run. Informed consent was obtained verbally in accordance with the Pennsylvania State University Institutional Review Board (Approval Number: STUDY00020301).

Subject demographics were collected, including age, sex, height, and weight. Running demographics regarding mileage, pacing, experience with a stroller, and comfort pushing a stroller were collected for each participant.

Blue Trident Inertial Measurement Unit (IMU) (Vicon, Centennial, CO) was taped to the skin at the anteromedial aspect of the distal tibia on the dominant limb and wrapped securely to minimize movement and bounce. Optical markers were placed on the sacrum and both feet to track foot strikes and speed. All subjects wore standardized lab shoes (Air Pegasus, Nike, Beaverton, OR).

### Experimental protocol

After IMU, camera, and subject marker calibration, subjects acclimated on a treadmill for a minimum of three minutes at their preferred running speed. Subjects then ran down the runway at their preferred overground speed while timing gates (Brower Timing Systems, Draper, UT) gathered their average speed across a minimum of five trials once their speed of choice had stabilized. These trials were not included in the analysis. For all later trials, subjects had to run within 10% of this average speed. If they did not, the trial was excluded and repeated. The presentation order for the stroller was randomized per subject. Rest was provided as needed between trials to minimize the effects of fatigue. Subjects ran across an 18 m runway with and without the running stroller over a force plate (Bertec 6090, Columbus, OH). Subjects pushed a double jogging stroller (Babytrend Expedition EX Double Jogger – Griffin), massing 14.22 kg and carrying a 9.28 kg mass (acting as a proxy baby) in the seat (23.5 kg total) with both hands. A double stroller was used to fully span the width of

the 60 cm force plate. The combined weight of the "baby" and stroller were comparable to that of a single stroller. Trials were removed and repeated if they were outside the speed bounds, the dominant foot did not cleanly strike the force plate, or the stroller wheels were on the force plate. A 12-camera Vicon (8 Bonita, 4 Vero) motion tracking system was used to track optical markers at 200 Hz. Subjects ran up to ten trials with the stroller and ten without. All force plate data were sampled at 2000 Hz and the IMUs at 400 Hz. Data collection, marker processing, and data synchronization were performed in Vicon Nexus (v2.15).

## Data pre-processing

All data pre-processing was performed in MATLAB (R2024b; MathWorks, Natick MA). The six channels of force plate data (three forces, three moments) were first trimmed by removing the first and last ten samples. Channels were then zeroed by subtracting the mean values of the last 51 samples, always after toe-off, on each channel. Each channel was then filtered using a zero-lag, 4th-order Butterworth lowpass filter with a 70 Hz cutoff frequency. The time of the footstrike was determined when the vertical ground reaction force (vGRF) was last above 0.001% BW before its maximum value. Toe-off time was determined when the vGRF was first below 0.001% BW after its maximum value. The data were then trimmed to this stance phase. All force data were normalized by the subject's body weight (BW), and all moment data were normalized by the product of the subject's body weight and stature (St).

The first- and second-time derivatives of the vGRF were calculated using 4th-order-accuracy central differencing. The vertical impact peak (VIP) was determined as the first local maximum of the vGRF before 50 ms [23]. For trajectories without a local maximum, an *effective* VIP was found by finding when the second derivative of vGRF first crosses zero from a negative value to a positive value. These effective VIP values are not included in VIP analysis, but their locations are used for subsequent calculations. The vertical propulsive peak (VPP) was calculated as the maximum value of vGRF after the VIP. The vertical instantaneous load rate (VILR) was calculated as the maximum value of the first derivative of vGRF between 20% and 80% of the VIP value [20]. The vertical average loading rate (VALR) was calculated as the slope from the linear regression of vGRF points between 20% and 80% of the VIP value. This regression was considered valid if $p < 0.05$ and $r^2 > 0.90$. The vertical impulse (VIMP) was calculated using numerical integration (trapz in MATLAB) of the vGRF over the stance phase.

The time of the zero-crossing for anterior-posterior (AP) direction was found as the first sign change from a negative value (braking) to a positive value (propulsion). The AP impulse (APIMP) was calculated by the net numerical integration of the AP force over stance phase.

Free moment (FM) calculations were performed using the methods and conventions defined previously [4]. Specifically, positive FM defines abduction for the right foot but defines adduction for the left foot. The absolute peak of the free moment ($|FM|_{max}$), peak adduction free moment (FMADD), and the free moment at peak braking force (FMBRAK) were extracted from the free moment curve. The integral or impulse of FM (FMIMP) was calculated using numerical integration over the stance phase.

IMU accelerometer data were filtered using a zero-lag, 4th-order, Butterworth low-pass filter with a 70 Hz cutoff frequency to match the processing done to the force plate data. The data were trimmed to the stance phase using the foot strike and toe-off times found from vGRF. The magnitude of acceleration was calculated using data from the three axes. The peak value of this magnitude over stance was extracted as resultant tibial acceleration (RTA) [24]. Magnitude values were used to nullify the effects of IMU orientation differences between subjects.

A subject's speed was calculated per trial using their sacrum marker's anterior displacement during one step. The non-dominant footstrike was determined using the minimum vertical height of the foot marker [25]. The anterior displacement of the sacrum during the period from non-dominant footstrike to dominant footstrike on the force plate was divided by the time during that period to calculate speed. In cases where this could not be calculated (eight trials) due to insufficient data prior to force plate contact, their speed determined from the speed gates was used.

## Statistical analysis

The mean value between conditions of all subjects' median variable of interest (VOI) values across up to ten trials were compared. Distributions of the differences between medians were generally not symmetric about the mean, not normally distributed, or contained outliers. Therefore, bootstrapping analysis [26] with a resample size of $10^6$ was used to determine the significance of the differences. The 5th and 95th percentiles of the mean of the differences from the bootstrapping were also reported. All VOI comparisons were performed using two-sided, paired-sample tests.

Effect size was calculated using Hedges's *g* [27,28]. Hedges's *g* was used over Cohen's *d* due to the number of subjects being fewer than 40. The effect size calculation was also performed using bootstrapping, with the 5th and 95th percentiles reported.

The mean speed for each subject between control and stroller running was compared. Here, two one-sided unpaired t-tests were used to determine how many subjects increased or decreased their speeds between conditions. Additionally, all trials among all subjects were grouped and the means compared between conditions using a two-sided unpaired t-test.

A significance level of $\alpha = 0.05$ was set *a priori* for all comparisons. All data processing and statistical analyses were performed in MATLAB.

## Results

38 runners were used for this analysis (Table 1). Subjects self-reported running 16.4 miles per week with a self-reported 5k time of 26 min. Twenty-seven (71.1%) of the runners identified as female. Fifteen subjects (39.5%) reported previous experience running with a stroller.

All following percent difference values indicate the mean within-subjects change for a particular VOI when running with a stroller compared to the control condition. When calculating vGRF parameters, 11.6% of trials did not have a clearly defined maximum that could be used for the VIP. Thus, the second derivative zero crossing was used to calculate loading rates. This affected all trials for only one subject; therefore, that subject was removed from only VIP analysis ($n = 37$). The mean change in vertical impact peak (VIP) and propulsive peak (VPP) decreased when running with a stroller by 16% and 10%, respectively (Fig 1, Table 2). The vertical impulse (VIMP) decreased by 9% with a stroller (Fig 1, Table 2). The vertical instantaneous loading rate (VILR) and average vertical loading rate (VALR) decreased when running with a stroller by 17% and 16%, respectively (Fig 2, Table 2). All VALR regression statistics fell within the preset $r^2$ and *p*-value criteria.

The braking peak (APBP) increased (*i.e.*, became more positive) by 24% in the anterior-posterior direction, indicating a smaller *magnitude* (Fig 3, Table 2). Propulsive peak (APPP) increased by 8%, indicating a larger magnitude (Fig. 3, Table 2). Both changes reflected forward acceleration, as demonstrated by the 350% increase in anterior-posterior impulse (APIMP) when running with a stroller (Fig 3, Table 2). This increase reflects a shift from near-zero APIMP in the control condition (–9.21E–4 BW*s) to 1.01E–2 BW*s with the stroller.

**Table 1. Summary statistics of $n = 38$ subjects reported as mean and standard deviation (S.D.).**

| Characteristic | Mean | S.D. |
|---|---|---|
| Stature (m) | 1.67 | 0.09 |
| Mass (kg) | 65.1 | 9.97 |
| Age (years) | 29.8 | 8.86 |
| Weekly running distance (miles) | 16.4 | 11.9 |
| 5k time (min) | 26.8 | 5.44 |
| Sex | 27 female (71.1%) | |
| Stroller running experience | 15 with experience (39.5%) | |

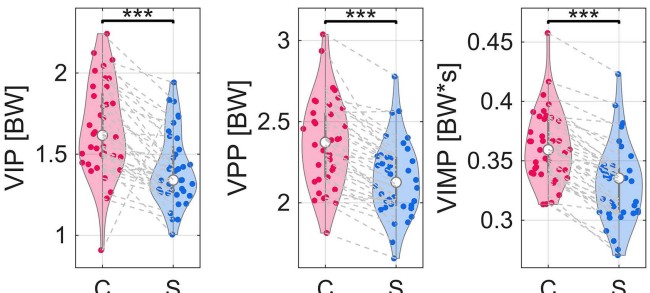

**Fig 1. Violin plots of distributions of impact peak (VIP), propulsive peak (VPP), and vertical impulse (VIMP) medians.** Gray dashed lines connect same subject between control (C) and stroller (S) conditions. One star indicates difference in medians with $p < 0.05$. Two stars indicate difference in medians with $p < 0.01$. Three stars indicate difference in medians with $p < 0.001$.

**Table 2. Summary statistics of variables of interest (VOI).** All *p*-values were determined using a paired, two-sample, two-sided, bootstrap with $10^6$ resamples. Mean Difference and Hedges's *g* effect size reported as [5th, 95th] percentile from bootstrapping. The units of *BW* stand for body weight, *St* for stature, and *g* for gravity.

| VOI | Control | Stroller | Mean Difference [5th, 95th] | *p*-value | Hedges's *g* [5th, 95th] | Mean % Difference |
|---|---|---|---|---|---|---|
| VIP [BW] | 1.65 | 1.41 | [-0.308, -0.171] | 0 | [0.481, 1.82] | −15.6 |
| VPP [BW] | 2.37 | 2.15 | [-0.257, -0.187] | 0 | [1.31, 2.35] | −9.69 |
| VIMP [BW*s] | 0.362 | 0.330 | [-0.0352, -0.0279] | 0 | [1.92, 2.76] | −9.22 |
| VILR [BW/s] | 90.2 | 75.8 | [-18.3, -10.5] | 0 | [0.702, 1.26] | −16.8 |
| VALR [BW/s] | 78.2 | 66.5 | [-15.0, -8.42] | 0 | [0.643, 1.28] | −16.1 |
| APBP [BW] | −0.362 | −0.283 | [0.0681, 0.089] | 0 | [1.59, 2.55] | +24.4 |
| APPP [BW] | 0.302 | 0.326 | [0.017, 0.0301] | 0 | [0.660, 1.31] | +7.88 |
| APIMP [BW*s] | −9.21E-4 | 1.01E-2 | [0.00989, 0.0122] | 0 | [2.09, 3.01] | +357 |
| FMADD [BW*St] | 6.37E-3 | 10.3E-3 | [0.00307, 0.00492] | 0 | [0.845, 1.64] | +46.4 |
| FMBRAK [BW*St] | 3.58E-3 | 5.70E-3 | [0.00133, 0.00293] | 0 | [0.454, 0.973] | +39.2 |
| |FM|max [BW*St] | 6.78E-3 | 10.4E-3 | [0.00271, 0.00452] | 0 | [0.787, 1.48] | +36.4 |
| FMIMP [BW*St*s] | 3.52E-4 | 9.80E-4 | [0.000487, 0.00078] | 0 | [0.905, 1.49] | +402 |
| RTA [g] | 7.70 | 7.19 | [-0.827, -0.181] | 0.0118 | [0.139, 0.800] | −7.27 |

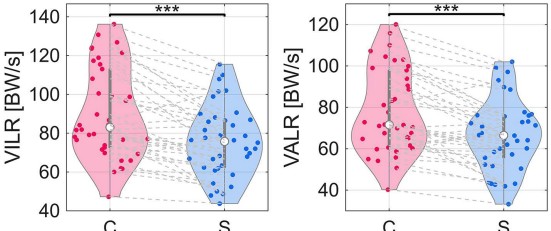

**Fig 2. Violin plots of distributions of vertical instantaneous loading rate (VILR) and vertical average (20-80%) loading rate (VALR) medians.** Gray dashed lines connect same subject between control (C) and stroller (S) conditions. One star indicates difference in medians with $p < 0.05$. Two stars indicate difference in medians with $p < 0.01$. Three stars indicate difference in medians with $p < 0.001$.

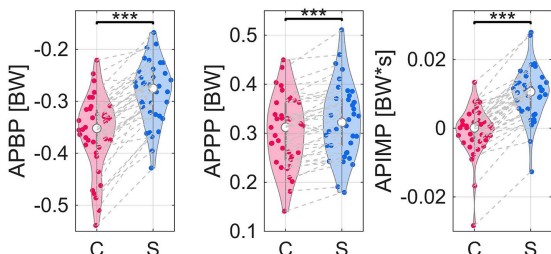

**Fig 3. Violin plots of distributions of peak braking force (APBP), propulsive force (APPP), and net impulse (APIMP) medians.** Gray dashed lines connect same subject between control (C) and stroller (S) conditions. One star indicates difference in medians with $p < 0.05$. Two stars indicate difference in medians with $p < 0.01$. Three stars indicate difference in medians with $p < 0.001$.

For the free moment, the adduction peak (FMADD), value at peak braking (FMBRAK), and absolute maximum ($|FM|_{max}$) all increased when running with a stroller by 46%, 39%, and 36%, respectively (Fig 4, Table 2). The free moment impulse (FMIMP) increased with a stroller by 402% (Fig 4, Table 2), representing an increase from nearly zero in the control condition (3.52E-4 BW*St*s) to a slight positive value for the stroller (9.80E-4 BW*St*s).

The tibial IMUs failed to collect data for two subjects properly and were not used for tibial acceleration analysis ($n = 36$). The peak resultant tibial acceleration (RTA) during stance decreased by 7% when running with a stroller (Fig 5, Table 2).

When comparing within each subject, nine subjects were found to have a significantly higher mean speed when running with the stroller compared to their control trials. In comparison, eleven subjects were found to have a significantly

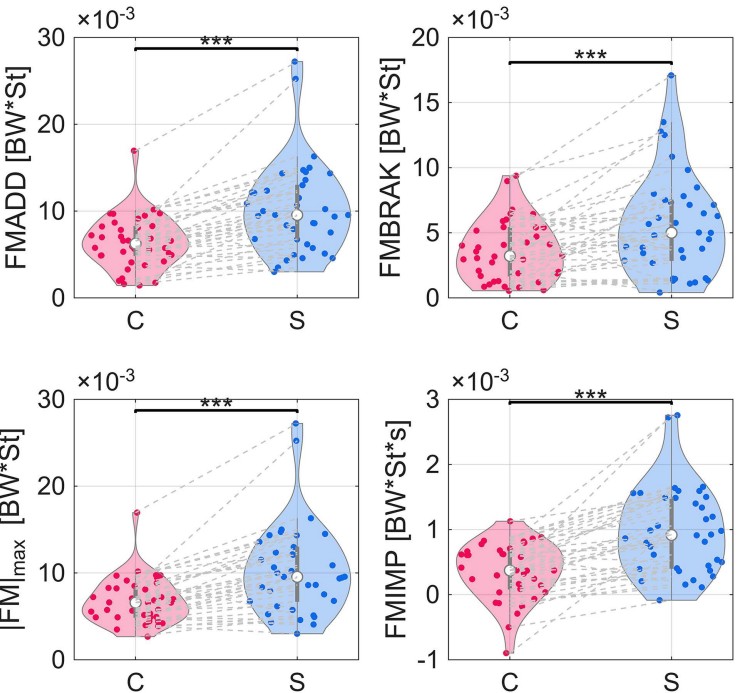

**Fig 4. Violin plots of distributions of peak free moment Adduction (FMADD), free moment at peak braking (FMBRAK), absolute maximum free moment value ($|FM|_{max}$), and net area under the free moment curve (FMIMP) medians.** All free moment values were normalized by the subject's bodyweight (BW) and stature (St). Gray dashed lines connect same subject between control (C) and stroller (S) conditions. One star indicates difference in medians with $p < 0.05$. Two stars indicate difference in medians with $p < 0.01$. Three stars indicate difference in medians with $p < 0.001$.

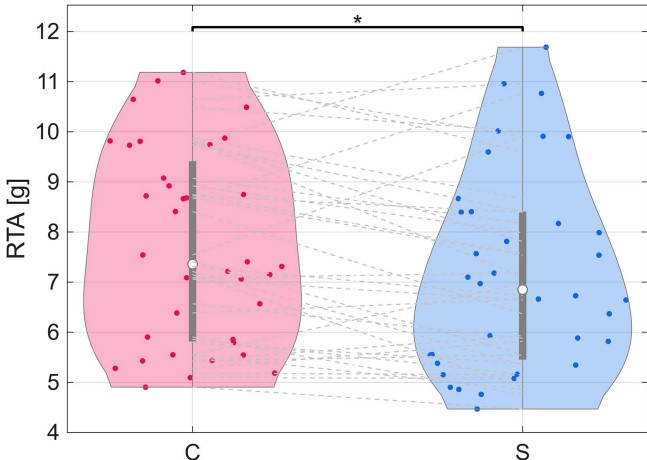

**Fig 5. Violin plots of distributions of peak resultant tibial acceleration (RTA) medians.** Gray dashed lines connect same subject between control (C) and stroller (S) conditions. One star indicates difference in medians with $p < 0.05$. Two stars indicate difference in medians with $p < 0.01$. Three stars indicate difference in medians with $p < 0.001$.

lower mean speed when running with the stroller. When all subjects were grouped, no significant difference was found between the mean speed for control ($3.29 \pm 0.469$ m/s) versus running with the stroller ($3.32 \pm 0.467$ m/s).

## Discussion

Generally, impact loading metrics were reduced, and torsional loading metrics were increased when running with a stroller, as was originally hypothesized. This suggests a potential tradeoff between a reduction in vertical force-related risk of injury, but an increase in torsional moment-related risk of injury while running with a stroller.

Vertical impact parameters, including VIP, VILR, VALR, and RTA, all decreased between 8–17% when running with a stroller. This reduction in impact loading may lead to a reduction in overuse injury risk [5,23,29–32]. In addition, lower VIMP has been associated with a decreased risk of bone stress injuries [33]. It is possible that runners changed their footstrike patterns to absorb impact load differently. Adopting a more anterior or extreme rearfoot strike pattern can mitigate impact loading [17]. Reductions in stride length have been shown to lead to reduced impact loading [18,34]. Most existing literature has shown that when speed is controlled, there is no change in stride length when running with a stroller [35–37]. However, one study showed that at faster speeds, stride length was reduced when running with a stroller, even when speed was matched [36]. Since this study utilized a slower, training pace, it is unlikely that vGRF parameters were affected by a stride length reduction alone.

Other possibilities must be considered to explain the reduction in impact loading. Due to the concomitant reduction in VPP, runners likely reduced underfoot loading by offloading force through the stroller's handlebars. This load-sharing mechanism likely reduced all vertical loading and RTA parameters. When pushing a stroller, runners tend to lean forward [37], which shifts their center of mass forward over the handlebars, supporting the notion that some force is redirected through the hands. A similar reduction in vertical load was observed when walking on the treadmill while holding the handlebars [38]. Taken together, the forward lean and handlebar evidence supports this load-sharing mechanism.

RTA was reduced proportionally less than the vGRF parameters when running with a stroller. These parameters have been shown to be related but not the same [39–41]. This mismatch in proportional reduction may be due to the location of the IMU on the shin rather than the direct measurement between the foot and the ground. In addition, acceleration from an IMU has a higher noise-to-signal ratio, potentially dampening the ability to observe a difference between control and stroller conditions.

The shift toward a more positive anterior-posterior ground reaction force and increase in anterior-posterior impulse indicates a runner with a positive horizontal acceleration while force data were measured. When runners pushed the stroller, it took them more distance to reach a constant speed. This lower acceleration to constant speed can be expected when pushing the additional weight of the stroller. As such, it is important to note that all forces with a stroller were measured during a period of acceleration (mean APIMP of 1.04E-2 BW*s), compared with a more constant speed (mean APIMP of –9.26E-4 BW*s) in the control condition. This positive acceleration may have also influenced other planes of force data. Extreme *deceleration* in sports movements has been shown to increase VIP [42–44]; thus, it is possible that the opposite is true, which is that acceleration decreases VIP. Compared to these studies examining deceleration, the acceleration rate in this study was much smaller. This indicates that the amount of acceleration occurring with the stroller will not likely cause a significant increase in impact parameters. Taken alone, the decreased magnitude of the braking force (APBRAK) may also be indicative of decreased running injury risk [39,40].

There is a relationship between speed and impact loading. As speed increases, VIP and VPP increase [41] as well as VALR and VILR [45–47]. Thus, it would be possible that the lower loading values with the stroller were due to lower speeds. However, since there was no significant difference observed in speed between conditions, and this effect was not observed in most participants, it is more likely that the increases in VIP and VPP were due to the stroller and not differences in speed.

All considered metrics of the torsional moment were increased when running with a stroller, which may be indicative of increased overuse injury risk [4,48]. While pushing the stroller, there is a need for increased torsional control to maintain the straightforward direction of the stroller, further increasing the free moment. Previous work has also shown an increase in free moment when carrying heavy loads [21]. Thus, the increased rotational inertia of the stroller may have further added to the increased free moment. It is also possible that this increased torsional load associated with the stroller may reflect a need to compensate for the lack of torsional control in the upper body. When running with the stroller, runners used both hands to push it, limiting their upper extremity rotation. It is possible that the restricted rotation from the stroller affects both arm swing and trunk rotation, which may be sufficient to influence the free moment under the foot. However, a small proof of concept study demonstrated that arm swing reduction alone has been shown to affect medial-lateral ground reaction force parameters and some kinematic parameters but had no effect on the free moment [49].

Future studies should examine the effect of reduced trunk rotation, direction control, and increased load on the free moment separately to determine the root cause of this increase. Alternative pushing styles, such as a one-handed push or the push and chase methods, should be considered [15]. One study has indicated that motion control footwear can help mitigate high free moment parameters [22]. Therefore, there may be a possibility that other mechanisms can be used to reduce this risk of injury when running with a stroller, such as reducing the weight or modifying the directional control mechanisms.

There were several limitations associated with this study. As previously mentioned, obtaining constant speed with a stroller was challenging along the lab runway. This was mitigated as much as possible by controlling speed between conditions and offering the maximal run-up that the space could provide. Future studies should examine ground reaction forces outside the laboratory to allow runners sufficient space to achieve constant speed. The use of a double stroller in this work may have exaggerated some of the findings, compared to using a single stroller; however, the weight was matched to be similar to that of a single stroller. Future work should examine the effect on biomechanics of a double- versus a single-stroller as differences in energetics were previously identified during walking [13]. This subject sample was a mixture of experienced and novice stroller runners who may have different adaptations to the stroller. Future studies should examine the effect of experience while pushing a stroller on their gait mechanics. The runner's height may have resulted in different pushing mechanics due to the placement of the handlebars and the footstrike space behind the stroller. The effects of participant height should be further examined in future studies. In addition, the runners used in this study were mostly female. While there is no evidence that males and females have different impact or free moment

characteristics [50], future studies should look to balance samples by sex to confirm. This study was well-controlled using a flat, straight, smooth indoor lab runway while minimizing the effects of fatigue. Real-world stroller running conditions often involve elevation changes, turns, and bumpier terrain than the smooth indoor runway. Future work should consider the effects of terrain on stroller running mechanics, as running uphill has been identified as more energetically demanding compared to flat terrain [14].

Despite these challenges, this was the first study to examine ground reaction force parameters when running with a stroller. Even when utilizing a conservative statistical design with bootstrapping and considering the lower bound of the effect size range, there were multiple statistically significant kinetic changes with moderate to large differences when running with a stroller. In general, a reduction in vertical force parameters may mitigate overuse injury risk, while increased torsional moment parameters may increase overuse injury risk. Future work should assess longitudinal injury rates in those running with strollers, kinematics associated with increased injury risk while running with a stroller, and the effect of terrain on stroller running mechanics. This work may be used to help runners, coaches, and stroller designers better understand the risks and challenges of stroller running.

## Acknowledgments

Subject data were collected with the aid of Faith Bentz, Naomi Fay, Matthew Holmes, Elizabeth McElroy, John Mora, Lauryn Morgan, Patrick O'Donnell, Colin Plunkett, Claire Samolewicz, and Douglas Wise.

## Author contributions

**Conceptualization:** Joseph M. Mahoney, Benjamin W. Infantolino, Allison R. Altman-Singles.

**Data curation:** Joseph M. Mahoney, Amy Lista, Diego Carbajal, Allison R. Altman-Singles.

**Formal analysis:** Joseph M. Mahoney, Allison R. Altman-Singles.

**Funding acquisition:** Amy Lista, Allison R. Altman-Singles.

**Investigation:** Joseph M. Mahoney, Diego Carbajal, Allison R. Altman-Singles.

**Methodology:** Joseph M. Mahoney, Amy Lista, Diego Carbajal, Benjamin W. Infantolino, Allison R. Altman-Singles.

**Project administration:** Joseph M. Mahoney, Amy Lista, Diego Carbajal, Allison R. Altman-Singles.

**Resources:** Joseph M. Mahoney, Benjamin W. Infantolino, Allison R. Altman-Singles.

**Software:** Joseph M. Mahoney, Allison R. Altman-Singles.

**Supervision:** Joseph M. Mahoney, Benjamin W. Infantolino, Allison R. Altman-Singles.

**Validation:** Joseph M. Mahoney, Benjamin W. Infantolino, Allison R. Altman-Singles.

**Visualization:** Joseph M. Mahoney, Allison R. Altman-Singles.

**Writing – original draft:** Joseph M. Mahoney, Diego Carbajal, Allison R. Altman-Singles.

**Writing – review & editing:** Joseph M. Mahoney, Amy Lista, Diego Carbajal, Benjamin W. Infantolino, Allison R. Altman-Singles.

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
