## [Decision Letter · Decision Letter 0]

4 Jun 2025

Dear Dr. Altman-Singles,

We look forward to receiving your revised manuscript.

Kind regards,

Laura-Anne Marie Furlong

Academic Editor

PLOS ONE

Journal Requirements:

“Funding for compensating research subjects and undergraduate research student stipends was provided by The Pennsylvania State University Berks College Research Development Grant, The Dr. Frank Franco Undergraduate Research Endowment, The Penn State Berks Cohen-Hammel Fellows Program, and Alvernia University Student Undergraduate Research Fellows  (SURF) Program. These funding sources were not involved in recruiting subjects, collecting or analyzing data, or writing and editing the manuscript.”

Reviewers' comments:

Reviewer's Responses to Questions

**Comments to the Author**

1. Is the manuscript technically sound, and do the data support the conclusions?

Reviewer #1: Yes

2. Has the statistical analysis been performed appropriately and rigorously?

Reviewer #1: Yes

3. Have the authors made all data underlying the findings in their manuscript fully available?

Reviewer #1: Yes

4. Is the manuscript presented in an intelligible fashion and written in standard English?

Reviewer #1: Yes

Reviewer #1: General Comments

I appreciate the opportunity to review the manuscript titled "Biomechanical Tradeoffs in Stroller Running: Reduced Vertical Impact Loading and Increased Torsional Injury Risk." The study addresses a relevant and underexplored topic with potential implications for injury prevention and running mechanics. Below, I offer several suggestions to enhance the manuscript's clarity, rigor, and impact.

Specific Comments

Introduction

1. The definition of a running stroller ("For the purposes of this analysis, a running stroller is defined as...") would be more appropriately placed in the Methods section under "Equipment." This adjustment would streamline the Introduction, allowing it to focus on broader context and avoid redundancy.

2. The Introduction could benefit from a more comprehensive review of recent literature on stroller running mechanics. Specifically, I recommend incorporating the following studies:

Sandbakk et al. (2020): Energetic Cost and Kinematics of Pushing a Stroller on Flat and Uphill Terrain (DOI: 10.3389/fphys.2020.00574).

Greany & Greany (2013): The Fitness Benefits of Pushing a Baby Stroller (DOI: 10.1097/JWH.0000000000000002).

These references would strengthen the theoretical foundation and highlight the practical relevance of the study.

3. The manuscript could further emphasize its practical implications by discussing how the findings might inform injury prevention strategies for caregivers who engage in stroller running. Additionally, the real-world applications for sports rehabilitation and running technique optimization could be articulated more clearly.

Methods

1. The manuscript currently describes participant demographics in the text but lacks a summary table. I strongly recommend including a table to present this information clearly and concisely.

2. The study exclusively uses the Babytrend Expedition EX Double Jogger. However, no rationale is provided for this choice. Could different stroller designs (e.g., single-wheel vs. double-wheel models) influence the results? A brief justification for the selected stroller would strengthen the methodological rigor.

3. While the study controls speed within ±10% variation, it does not address potential fatigue effects across multiple trials. Consider incorporating a post-trial Rating of Perceived Exertion (RPE) assessment or discussing fatigue as a potential limitation.

4. Given the use of an 18m indoor runway, it is unclear whether participants reached steady-state speed before force plate measurements. How was acceleration controlled? If speed varied significantly across trials, were any trials excluded based on extreme deviations? Clarifying these methodological details would enhance the reliability of the findings.

Results

1. In Table 1, I recommend recalculating APIMP and FMIMP to verify whether the observed extreme changes represent genuine effects or potential artifacts. This step would ensure the robustness of the reported findings.

2. To improve the interpretability of the results, I suggest including 95% confidence intervals in the bootstrap analysis. This addition would provide a more reliable estimation of variability.

3. To complement statistical significance testing, I recommend calculating Cohen’s d to assess the practical significance of the observed differences. This would help contextualize the real-world relevance of the findings.

4. Please revise the p-value formatting to avoid expressions such as "p = 0." Instead, use "p < 0.001" or report the exact p-values where appropriate to maintain precision.

5. Table 1 would benefit from clearer units and labels (e.g., BW, St) to ensure consistency and improve readability.

Discussion

1. The Discussion suggests that stroller running reduces vertical impact forces due to load-sharing through the arms. However, the extent of force offloading to the upper body remains unclear. Has prior research quantified hand force contributions during stroller running or similar load-assisted activities (e.g., trekking poles, exoskeletons)? Addressing this would provide a more nuanced understanding of the findings.

2. The study proposes that limited upper-body rotation leads to increased free moment and torsional stress. What biomechanical evidence supports this causal relationship? Expanding on this mechanism would strengthen the Discussion.

3. The first paragraph of the Discussion mentions stride length differences but does not clearly explain or resolve these discrepancies. A more detailed analysis or framework for interpreting these differences would be valuable.

4. The study reports significant changes in impact forces and torsional loads despite unchanged running speed. How can these findings be explained? Could postural adjustments (e.g., increased forward lean) or changes in muscle activation patterns influence force distribution, even at constant speeds? Exploring these possibilities would deepen the Discussion.

**Do you want your identity to be public for this peer review?** For information about this choice, including consent withdrawal, please see our Privacy Policy

Reviewer #1: No

---

## [Author Response · Author response to Decision Letter 1]

24 Jun 2025

(***Also uploaded as an attachment***)

Author Comments: We would like to thank the Reviewer for their thoughtful and constructive feedback. Your careful reading and detailed suggestions have helped us strengthen the clarity, methodological rigor, and overall presentation of our manuscript. In the responses below, we address each of your comments point by point and describe the specific revisions made in the manuscript. Where we chose an alternative approach, we provide a rationale to clarify our decisions. We appreciate the opportunity to improve the work and believe the manuscript is stronger as a result.

Introduction

1. The definition of a running stroller ("For the purposes of this analysis, a running stroller is defined as...") would be more appropriately placed in the Methods section under "Equipment." This adjustment would streamline the Introduction, allowing it to focus on broader context and avoid redundancy.

Author Response: We revised the statement to improve clarity but chose to keep it in the Introduction. Its purpose is to narrow the scope of vehicles considered in the literature review and discussion. Placing it here helps frame the context early on and ensures readers understand the specific type of stroller being referenced throughout the manuscript.

2. The Introduction could benefit from a more comprehensive review of recent literature on stroller running mechanics. Specifically, I recommend incorporating the following studies:

Sandbakk et al. (2020): Energetic Cost and Kinematics of Pushing a Stroller on Flat and Uphill Terrain (DOI: 10.3389/fphys.2020.00574).

Greany & Greany (2013): The Fitness Benefits of Pushing a Baby Stroller (DOI: 10.1097/JWH.0000000000000002).

These references would strengthen the theoretical foundation and highlight the practical relevance of the study.

Author Response: Thank you for these suggestions. Both papers have been reviewed and integrated into the Introduction to provide additional context and support for the need to study the biomechanics of stroller running, particularly in contrast to the energetic and fitness outcomes reported in those works. We also reference these studies in the Discussion to help motivate future directions, including biomechanical differences in uphill and downhill running, and comparisons between single- and double-stroller conditions.

3. The manuscript could further emphasize its practical implications by discussing how the findings might inform injury prevention strategies for caregivers who engage in stroller running. Additionally, the real-world applications for sports rehabilitation and running technique optimization could be articulated more clearly.

Author Response: We’ve expanded the end of the Introduction to better highlight these practical applications. We now discuss how our findings may inform injury prevention strategies, especially in relation to increased free moment. We also mention motion control footwear as one possible intervention. These insights could have implications not only for recreational caregivers but also for broader applications in rehabilitation and running technique refinement.

Methods

1. The manuscript currently describes participant demographics in the text but lacks a summary table. I strongly recommend including a table to present this information clearly and concisely.

Author Response: Participant demographics have been added as Table 1 in the Results section to provide a clearer and more concise summary of this information.

2. The study exclusively uses the Babytrend Expedition EX Double Jogger. However, no rationale is provided for this choice. Could different stroller designs (e.g., single-wheel vs. double-wheel models) influence the results? A brief justification for the selected stroller would strengthen the methodological rigor.

Author Response: The Babytrend Expedition EX Double Jogger was selected for its wide wheelbase, which allowed the rear wheels to clear the force plate and ensured accurate data collection. We’ve added this explanation to the Experimental Protocol section.

We also address the use of a double stroller in the Limitations subsection of the Discussion. Ongoing work in our lab is examining how biomechanics differ between single and double stroller designs, which we note as a direction for future research.

3. While the study controls speed within ±10% variation, it does not address potential fatigue effects across multiple trials. Consider incorporating a post-trial Rating of Perceived Exertion (RPE) assessment or discussing fatigue as a potential limitation.

Author Response: Fatigue effects were mitigated by counterbalancing the order of conditions and allowing participants to rest as much as needed between trials. This was part of the study design and has now been added to the Experimental Protocol section. We consider this a strength of the protocol, as it helped ensure consistency across conditions, and have clarified that in the Discussion. While we did not include RPE in this protocol, we agree that it could be informative and are incorporating it in our follow-up studies

4. Given the use of an 18m indoor runway, it is unclear whether participants reached steady-state speed before force plate measurements. How was acceleration controlled? If speed varied significantly across trials, were any trials excluded based on extreme deviations? Clarifying these methodological details would enhance the reliability of the findings.

Author Response: This concern is valid and has been addressed in two places. In the Discussion, we note that most participants were still accelerating during the stroller trials and that acceleration was not controlled in the protocol. In the Methods section, we’ve clarified that any trials with speeds outside ±10% of a participant’s average were excluded from analysis.

Results

1. In Table 1, I recommend recalculating APIMP and FMIMP to verify whether the observed extreme changes represent genuine effects or potential artifacts. This step would ensure the robustness of the reported findings.

Author Response: This is now Table 2 in the text.

The extreme percent differences in APIMP and FMIMP are genuine and reflect the fact that control condition values are near zero, which inflates the relative change. To improve interpretability, we’ve added the 5th and 95th percentiles to show the absolute differences and give more context to the reported values.

2. To improve the interpretability of the results, I suggest including 95% confidence intervals in the bootstrap analysis. This addition would provide a more reliable estimation of variability.

Author Response: We’ve added 95% confidence intervals to the bootstrapped estimates in Table 2. This addition strengthens the interpretation of the observed differences and directly supports the context we’ve added in response to item 1.

3. To complement statistical significance testing, I recommend calculating Cohen’s d to assess the practical significance of the observed differences. This would help contextualize the real-world relevance of the findings.

Author Response: We’ve incorporated effect size using Hedges’s g, which is more appropriate for small sample sizes. These values are included in Table 2 along with the 5th and 95th percentile bounds from bootstrapping. We also added a brief discussion of effect sizes at the end of the Discussion section to help contextualize the practical relevance of the findings alongside statistical significance.

4. Please revise the p-value formatting to avoid expressions such as "p = 0." Instead, use "p < 0.001" or report the exact p-values where appropriate to maintain precision.

Author Response: The p-values in Table 2 were calculated using bootstrapping, and "p = 0" reflects the direct output of that method. We’ve addressed this concern by providing 95% confidence intervals (as noted above in item 2), which offer a more meaningful sense of variability and effect strength.

5. Table 1 would benefit from clearer units and labels (e.g., BW, St) to ensure consistency and improve readability.

Author Response: This is now Table 2.

Units are listed in the first column next to each metric, and we’ve clarified all abbreviations and units in the table caption to improve readability and consistency across the table.

Discussion

1. The Discussion suggests that stroller running reduces vertical impact forces due to load-sharing through the arms. However, the extent of force offloading to the upper body remains unclear. Has prior research quantified hand force contributions during stroller running or similar load-assisted activities (e.g., trekking poles, exoskeletons)? Addressing this would provide a more nuanced understanding of the findings.

Author Response: We’ve added a reference to a study where participants used a handrail while walking on a treadmill and showed a decrease in vertical ground reaction forces. This supports the idea that some impact force can be offloaded through the arms.

Studies involving trekking poles had additional complications, such as sloped surfaces, and often didn’t measure vGRF, so they don’t align well with our current setup. That said, they will be useful for informing our future work on uphill and downhill running.

2. The study proposes that limited upper-body rotation leads to increased free moment and torsional stress. What biomechanical evidence supports this causal relationship? Expanding on this mechanism would strengthen the Discussion.

Author Response: We have revised the paragraph based on preliminary data from our ongoing pilot study, which isolates the effects of trunk rotation, arm swing, and steering on the free moment (FM). Our findings so far align with the results from the Miller paper, which did not identify arm swing as a significant contributor to changes in FM.

In the revised version, we begin by discussing increased rotational inertia as a potential contributing factor. We then address the roles of arm swing and trunk rotation, suggesting that their absence is less likely to be the primary cause of increased FM.

3. The first paragraph of the Discussion mentions stride length differences but does not clearly explain or resolve these discrepancies. A more detailed analysis or framework for interpreting these differences would be valuable.

Author Response: This paragraph has been rewritten for clarity. In previous studies, stride length did not change when runners had constant speed between control and stroller running. We present this to show that a change in stride length is likely not the reason for changes in the loading or FM. The step length was not presented in this paper, but will be included in a forthcoming paper focusing on the kinetics from this experiment.

4. The study reports significant changes in impact forces and torsional loads despite unchanged running speed. How can these findings be explained? Could postural adjustments (e.g., increased forward lean) or changes in muscle activation patterns influence force distribution, even at constant speeds? Exploring these possibilities would deepen the Discussion.

Author Response: The most likely explanation for the observed reductions in impact forces and torsional loads, despite unchanged running speed, is a redistribution of load from the runner to the stroller. This interpretation is supported by the newly added reference on handrail use during treadmill walking, which demonstrated a comparable offloading effect on vertical ground reaction forces.

This redistribution may be partially explained by an increase in forward trunk lean, as documented in O’Sullivan et al., 2016. We have collected additional data related to this hypothesis that are still being processed and expect these findings will help clarify the underlying mechanisms. We have added text to the Discussion describing this possibility.

Although we did not collect muscle activation data in this study, we agree that changes in neuromuscular strategy may also contribute. We expect that future work in our lab will examine both muscular and kinematic adjustments in greater detail. These directions have been noted in the revised Discussion to clarify the scope of the current analysis and to outline potential next steps.

---

## [Decision Letter · Decision Letter 1]

3 Sep 2025

Biomechanical Tradeoffs in Stroller Running: Reduced Vertical Impact Loading and Increased Torsional Injury Risk

PONE-D-25-04144R1

Dear Dr. Altman-Singles,

We’re pleased to inform you that your manuscript has been judged scientifically suitable for publication and will be formally accepted for publication once it meets all outstanding technical requirements.

Kind regards,

Laura-Anne Marie Furlong

Academic Editor

PLOS ONE

Additional Editor Comments (optional):

Reviewer #1:

Reviewers' comments:

Reviewer's Responses to Questions

**Comments to the Author**

Reviewer #1: All comments have been addressed

2. Is the manuscript technically sound, and do the data support the conclusions?

Reviewer #1: Yes

3. Has the statistical analysis been performed appropriately and rigorously?

Reviewer #1: Yes

4. Have the authors made all data underlying the findings in their manuscript fully available?

Reviewer #1: Yes

5. Is the manuscript presented in an intelligible fashion and written in standard English?

Reviewer #1: Yes

Reviewer #1: I have reviewed the authors’ responses and the revised manuscript. They have addressed the main issues previously raised, including the addition of the demographic table, supplementary results, updated references, and revisions to the Discussion section. Most of the suggestions have been implemented in the manuscript, and for the one or two points not modified, the authors have provided reasonable explanations. Overall, I find the revisions satisfactory and have no major concerns.

**Do you want your identity to be public for this peer review?** For information about this choice, including consent withdrawal, please see our Privacy Policy

Reviewer #1: No

---

## [Editor Report · Acceptance letter]

PONE-D-25-04144R1

PLOS ONE

Dear Dr. Altman-Singles,

I'm pleased to inform you that your manuscript has been deemed suitable for publication in PLOS ONE. Congratulations! Your manuscript is now being handed over to our production team.

Kind regards,

on behalf of

Dr. Laura-Anne Marie Furlong

Academic Editor

PLOS ONE